# DyNet: Dynamic Convolution for Accelerating Convolution Neural Networks

## Abstract

Convolution operator is the core of convolutional neural networks (CNNs) and occupies the most computation cost. To make CNNs more efficient, many methods have been proposed to either design lightweight networks or compress models. Although some efficient network structures have been proposed, such as MobileNet or ShuffleNet, we find that there still exists redundant information between convolution kernels. To address this issue, we propose a novel dynamic convolution method named **DyNet** in this paper, which can adaptively generate convolution kernels based on image contents. To demonstrate the effectiveness, we apply DyNet on multiple state-of-the-art CNNs. The experiment results show that DyNet can reduce the computation cost remarkably, while maintaining the performance nearly unchanged. Specifically, for ShuffleNetV2 (1.0), MobileNetV2 (1.0), ResNet18 and ResNet50, DyNet reduces 37.0%, 54.7%, 67.2% and 71.3% FLOPs respectively while the Top-1 accuracy on ImageNet only changes by $+1.0\%$, $-0.27\%$, $-0.6\%$ and $-0.08\%$. Meanwhile, DyNet further accelerates the inference speed of MobileNetV2 (1.0), ResNet18 and ResNet50 by $1.87\times$, $1.32\times$ and $1.48\times$ on CPU platform respectively. To verify the scalability, we also apply DyNet on segmentation task, the results show that DyNet can reduces 69.3% FLOPs while maintaining the Mean IoU on segmentation task.

## 1 Introduction

Convolutional neural networks (CNNs) have achieved state-of-the-art performance in many computer vision tasks (Krizhevsky et al., 2012; Szegedy et al., 2013), and the neural architectures of CNNs are evolving over the years (Krizhevsky et al., 2012; Simonyan & Zisserman, 2014; Szegedy et al., 2015; He et al., 2016; Hu et al., 2018; Zhong et al., 2018a;b). However, modern high-performance CNNs often require a lot of computation resources to execute large amount of convolution kernel operations. Aside from the accuracy, to make CNNs applicable on mobile devices, building lightweight and efficient deep models has attracting much more attention recently (Howard et al., 2017; Sandler et al., 2018; Zhang et al., 2018; Ma et al., 2018). These methods can be roughly categorized into two types: efficient network design and model compression. Representative methods for the former category are MobileNet (Howard et al., 2017; Sandler et al., 2018) and ShuffleNet (Ma et al., 2018; Zhang et al., 2018), which use depth-wise separable convolution and channel-level shuffle techniques to reduce computation cost. On the other hand, model compression based methods tend to obtain a smaller network by compressing a larger network via pruning, factorization or mimic (Chen et al., 2015; Han et al., 2015a; Jaderberg et al., 2014; Lebedev et al., 2014; Ba & Caruana, 2014).

Although some handcrafted efficient network structures have been designed, we observe that the significant correlations still exist among convolutional kernels, and introduce large amount of redundant calculations. Moreover, these small networks are hard to compress. For example, Liu et al. (2019) compress MobileNetV2 to 124M, but the accuracy drops by $5.4\%$ on ImageNet. We theoretically analyze above observation, and find that this phenomenon is caused by the nature of static convolution, where correlated kernels are cooperated to extract noise-irrelevant features. Thus it is hard to compress the fixed convolution kernels without information loss. We also find that if we linearly fuse several convolution kernels to generate one dynamic kernel based on the input, we can obtain the noise-irrelevant features without the cooperation of multiple kernels, and further reduce the computation cost of convolution layer remarkably.

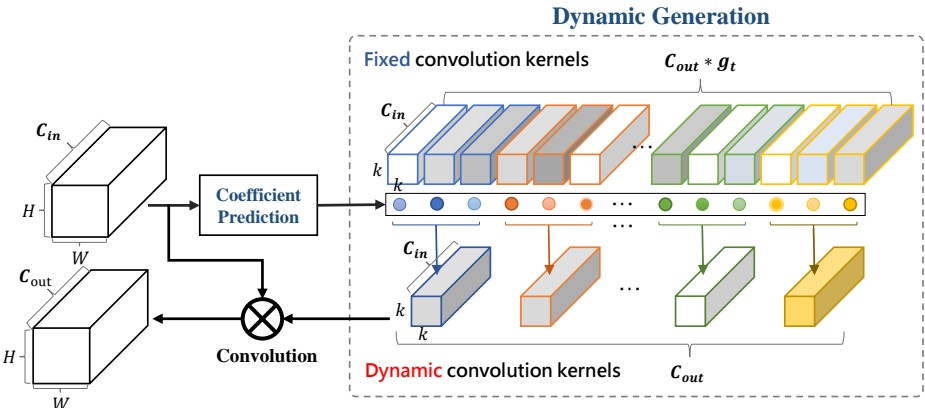

Figure 1: The overall framework of the proposed dynamic convolution.

Based on above observations and analysis, in this paper, we propose a novel dynamic convolution method named DyNet. The overall framework of DyNet is shown in Figure 1, which consists of a *coefficient prediction module* and a *dynamic generation module*. The coefficient prediction module is trainable and designed to predict the coefficients of fixed convolution kernels. Then the dynamic generation module further generates a dynamic kernel based on the predicted coefficients.

Our proposed dynamic convolution method is simple to implement, and can be used as a drop-in plugin for any convolution layer to reduce computation cost. We evaluate the proposed DyNet on state-of-the-art networks such as MobileNetV2, ShuffleNetV2 and ResNets. Experiment results show that DyNet reduces $37.0\%$ FLOPs of ShuffleNetV2 (1.0) while further improve the Top-1 accuracy on ImageNet by $1.0\%$. For MobileNetV2 (1.0), ResNet18 and ResNet50, DyNet reduces $54.7\%$, $67.2\%$ and $71.3\%$ FLOPs respectively, the Top-1 accuracy on ImageNet changes by $-0.27\%$, $-0.6\%$ and $-0.08\%$. Meanwhile, DyNet further accelerates the inference speed of MobileNetV2 (1.0), ResNet18 and ResNet50 by $1.87\times$, $1.32\times$ and $1.48\times$ on CPU platform respectively.

## 2 RELATED WORK

We review related works from three aspects: efficient convolution neural network design, model compression and dynamic convolutional kernels.

### 2.1 EFFICIENT CONVOLUTION NEURAL NETWORK DESIGN

In many computer vision tasks (Krizhevsky et al., 2012; Szegedy et al., 2013), model design plays a key role. The increasing demands of high quality networks on mobile/embedding devices have driven the study on efficient network design (He & Sun, 2015). For example, GoogleNet (Szegedy et al., 2015) increases the depth of networks with lower complexity compared to simply stacking convolution layers; SqueezeNet (Iandola et al., 2016) deploys a bottleneck approach to design a very small network; Xception (Chollet, 2017), MobileNet (Howard et al., 2017) and MobileNetV2 (Sandler et al., 2018) use depth-wise separable convolution to reduce computation and model size. ShuffleNet (Zhang et al., 2018) and ShuffleNetV2 (Ma et al., 2018) shuffle channels to reduce computation of $1 \times 1$ convolution kernel and improve accuracy. Despite the progress made by these efforts, we find that there still exists redundancy between convolution kernels and cause redundant computation.

### 2.2 MODEL COMPRESSION

Another trend to obtaining small network is model compression. Factorization based methods (Jaderberg et al., 2014; Lebedev et al., 2014) try to speed up convolution operation by using tensor decomposition to approximate original convolution operation. Knowledge distillation based methods (Ba & Caruana, 2014; Romero et al., 2014; Hinton et al., 2015) learn a small network to mimic a larger teacher network. Pruning based methods (Han et al., 2015a;b; Wen et al., 2016; Liu et al., 2019)

try to reduce computation by pruning the redundant connections or convolution channels. Compared with those methods, DyNet is more effective especially when the target network is already efficient enough. For example, in (Liu et al., 2019), they get a smaller model of 124M FLOPs by pruning the MobileNetV2, however it drops the accuracy by $5.4\%$ on ImageNet compared with the model with 291M FLOPs. While in DyNet, we can reduce the FLOPs of MobileNetV2 (1.0) from 298M to 129M with the accuracy drops only $0.27\%$.

### 2.3 DYNAMIC CONVOLUTION KERNEL

Generating dynamic convolution kernels appears in both computer vision and natural language processing (NLP) tasks.

In computer vision domain, Klein et al. (Klein et al., 2015) and Brabandere et al. (Jia et al., 2016) directly generate convolution kernels via a linear layer based on the feature maps of previous layers. Because convolution kernels has a large amount of parameters, the linear layer will be inefficient on the hardware. Our proposed method solves this problem via merely predicting the coefficients for linearly combining static kernels and achieve real speed up for CNN on hardware. The idea of linearly combining static kernels using predicted coefficients has been proposed by Yang et al. (Yang et al., 2019), but they focus on using more parameters to make models more expressive while we focus on reducing redundant calculations in convolution. We make theoretical analysis and conduct correlation experiment to prove that correlations among convolutional kernels can be reduced via dynamically fusing several kernels.

In NLP domain, some works (Shen et al., 2018; Wu et al., 2019; Gong et al., 2018) incorporate context information to generate input-aware convolution filters which can be changed according to input sentences with various lengths. These methods also directly generate convolution kernels via a linear layer, etc. Because the size of CNN in NLP is smaller and the dimension of convolution kernel is one, the inefficiency issue for the linear layer is alleviated. Moreover, Wu et al. (Wu et al., 2019) alleviate this issue utilizing the depthwise convolution and the strategy of sharing weight across layers. These methods are designed to improve the adaptivity and flexibility of language modeling, while our method aims to cut down the redundant computation cost.

## 3 DYNET: DYNAMIC CONVOLUTION IN CNNS

In this section, we first describe the motivation of DyNet. Then we explain the proposed dynamic convolution in detail. Finally, we illustrate the DyNet based architectures of our proposed Dy-mobile, Dy-shuffle, Dy-ResNet18, Dy-ResNet50.

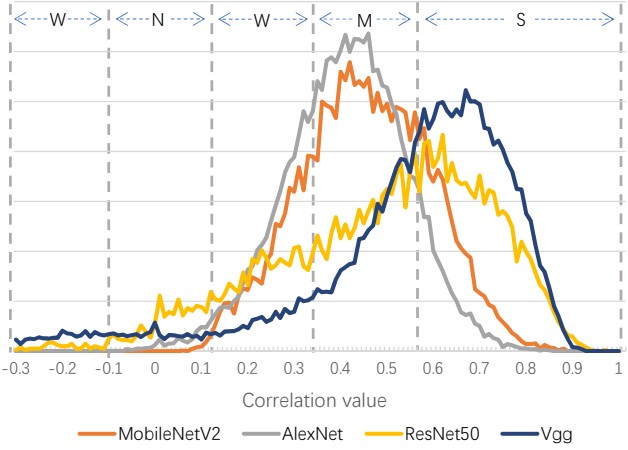

Figure 2: Pearson product-moment correlation coefficient between feature maps. S, M, W, N denote strong, middle, weak and no correlation respectively.

## 3.1 MOTIVATION

As illustrated in previous works (Han et al., 2015a;b; Wen et al., 2016; Liu et al., 2019), convolutional kernels are naturally correlated in deep models. For some of the well known networks, we plot the distribution of Pearson product-moment correlation coefficient between feature maps in Figure 2. Most existing works try to reduce correlations by compressing. However, efficient and small networks like MobileNets are harder to prune despite the correlation is still significant. We think these correlations are vital for maintaining the performance because they are cooperated to obtain noise-irrelevant features. We take face recognition as an example, where the pose or the illumination is not supposed to change the classification results. Therefore, the feature maps will gradually become noise-irrelevant when they go deeper. Based on the theoretical analysis in appendix A, we find that if we dynamically fuse several kernels, we can get noise-irrelevant feature without the cooperation of redundant kernels. In this paper, we propose dynamic convolution method, which learns the coefficients to fuse multiple kernels into a dynamic one based on image contents. We give more in depth analysis about our motivation in appendix A.

## 3.2 DYNAMIC CONVOLUTION

The goal of dynamic convolution is to learn a group of kernel coefficients, which fuse multiple fixed kernels to a dynamic one. We demonstrate the overall framework of dynamic convolution in Figure 1. We first utilize a trainable coefficient prediction module to predict coefficients. Then we further propose a dynamic generation module to fuse fixed kernels to a dynamic one. We will illustrate the coefficient prediction module and dynamic generation module in detail in the following of this section.

**Coefficient prediction module** Coefficient prediction module is proposed to predict coefficients based on image contents. As shown in Figure 3, the coefficient prediction module can be composed by a global average pooling layer and a fully connected layer with Sigmoid as activation function. Global average pooling layer aggregates the input feature maps into a $1 \times 1 \times C_{in}$ vector, which serves as a feature extraction layer. Then the fully connected layer further maps the feature into a $1 \times 1 \times C$ vector, which are the coefficients for fixed convolution kernels of several dynamic convolution layers.

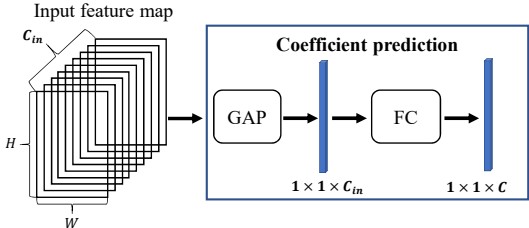

Figure 3: The coefficient prediction module.

**Dynamic generation module** For a dynamic convolution layer with weight $[C_{out} \times g_t, C_{in}, k, k]$, it corresponds with $C_{out} \times g_t$ fixed kernels and $C_{out}$ dynamic kernels, the shape of each kernel is $[C_{in}, k, k]$. $g_t$ denotes the group size, it is a hyperparameter. We denote the fixed kernels as $w_t^i$, the dynamic kernels as $\widetilde{w_t}$, the coefficients as $\eta_t^i$, where $t = 0, ..., C_{out}, i = 0, ..., g_t$.

After the coefficients are obtained, we generate dynamic kernels as follows:

$$\widetilde{w}_t = \sum_{i=1}^{g_t} \eta_t^i \cdot w_t^i \tag{1}$$

**Training algorithm** For the training of the proposed dynamic convolution, it is not suitable to use batch based training scheme. It is because the convolution kernel is different for different input images in the same mini-batch. Therefore, we fuse feature maps based on the coefficients rather than kernels during training. They are mathematically equivalent as shown in Eq. 2:

$$\widetilde{O}_t = \widetilde{w}_t \otimes x = \sum_{i=1}^{g_t} (\eta_t^i \cdot w_t^i) \otimes x = \sum_{i=1}^{g_t} (\eta_t^i \cdot w_t^i \otimes x)$$

$$= \sum_{i=1}^{g_t} (\eta_t^i \cdot (w_t^i \otimes x)) = \sum_{i=1}^{g_t} (\eta_t^i \cdot O_t^i), \tag{2}$$

where $x$ denotes the input, $\widetilde{O}_t$ denotes the output of dynamic kernel $\widetilde{w}_t$, $O_t^i$ denotes the output of fixed kernel $w_t^i$.

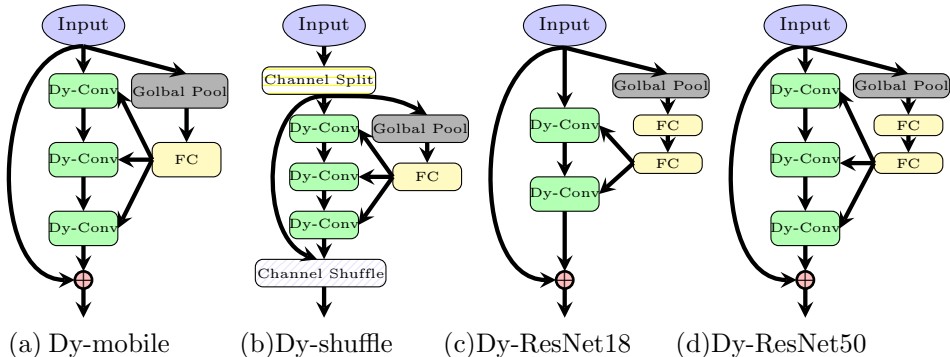

(a) Dy-mobile     (b)Dy-shuffle     (c)Dy-ResNet18     (d)Dy-ResNet50

Figure 4: Basic building bolcks for Dynamic Network variants of MobileNet (a), shuffleNet (b)?ResNet18 (c), and ResNet50 (d).

### 3.3 DYNAMIC CONVOLUTION NEURAL NETWORKS

We equip the MobileNetV2, ShuffleNetV2 and ResNets with our proposed dynamic convolution, and propose Dy-mobile, Dy-shuffle, Dy-ResNet18 and Dy-ResNet50 respectively. The building blocks of these 4 network are shown in Figure 4. Based on above dynamic convolution, each dynamic kernel can get noise-irrelevant feature without the cooperation of other kernels. Therefore we can reduce the channels for each layer of those base models and remain the performance. We set the hyper-parameter $g_t$ as 6 for all of them, and we give details of these dynamic CNNs below.

**Dy-mobile** In our proposed Dy-mobile, we replace the original MobileNetV2 block with our dy-mobile block, which is shown in Figure 4 (a). The input of coefficient prediction module is the input of block, it produces the coefficients for all three dynamic convolution layers. Moreover, we further make two adjustments:

- We do not expand the channels in the middle layer like MobileNetV2. If we denote the output channels of the block as $C_{out}$, then the channels of all the three convolution layers will be $C_{out}$.
- Since the depth-wise convolution is efficient, we set $groups = \frac{C_{out}}{6}$ for the dynamic depth-wise convolution. We will enlarge $C_{out}$ to make it becomes the multiple of 6 if needed.

After the aforementioned adjustments, the first dynamic convolution layer reduce the FLOPs from $6C^2HW$ to $C^2HW$. The second dynamic convolution layer keep the FLOPs as $6CHW \times 3^2$ unchanged because we reduce the output channels by 6x while setting the groups of convolution 6x smaller, too. For the third dynamic convolution layer, we reduce the FLOPs from $6C^2HW$ to $C^2HW$ as well. The ratio of FLOPs for the original block and our dy-mobile block is:

$$\frac{6C^2HW + 6CHW \times 3^2 + 6C^2HW}{C^2HW + 6CHW \times 3^2 + C^2HW} = \frac{6C + 27}{C + 27} = 6 - \frac{135}{C + 27} \tag{3}$$

**Dy-shuffle** In the original ShuffleNet V2, channel split operation will split feature maps to right-branch and left-branch, the right branch will go through one pointwise convolution, one depthwise convolution and one pointwise convolution sequentially. We replace conventional convolution with dynamic convolution in the right branch as shown in Figure 4 (b). We feed the input of right branch into coefficient prediction module to produce the coefficients. In our dy-shuffle block, we split channels into left-branch and right-branch with ratio 3 : 1, thus we reduce the 75% computation cost for two dynamic pointwise convolutuon. Similar with dy-mobile, we adjust the parameter "groups" in dynamic depthwise convolution to keep the FLOPs unchanged.

**Dy-ResNet18/50** In Dy-ResNet18 and DyResNet50, we simple reduce half of the output channels for dynamic convolution layers of each residual block. Because the input channels of each block is large compared with dy-mobile and dy-shuffle, we use two linear layer as shown in Figure 4 (c) and Figure 4 (d) to reduce the amount of parameters. If the input channel is $C_{in}$, the output channels of the first linear layer will be $\frac{C_{in}}{4}$ for Dy-ResNet18/50.

## 4 EXPERIMENTS

### 4.1 IMPLEMENTATION DETAILS

For the training of the proposed dynamic neural networks. Each image has data augmentation of randomly cropping and flipping, and is optimized with SGD strategy with cosine learning rate decay. We set batch size, initial learning rate, weight decay and momentum as 2048, 0.8, 5e-5 and 0.9 respectively. We also use the label smoothing with rate 0.1. We evaluate the accuracy on the test images with center crop.

### 4.2 EXPERIMENT SETTINGS AND COMPARED METHODS

We evaluate DyNet on ImageNet (Russakovsky et al., 2015), which contains 1.28 million training images and 50K validation images collected from 1000 different classes. We train the proposed networks on the training set and report the top-1 error on the validation set. To demonstrate the effectiveness, we compare the proposed dynamic convolution with state-of-the-art networks under mobile setting, including MobileNetV1 (Howard et al., 2017), MobileNetV2 (Sandler et al., 2018), ShuffleNet (Zhang et al., 2018), ShuffleNet V2 (Ma et al., 2018), Xception (Chollet, 2017), DenseNet (Huang et al., 2017), IGCV2 (Xie et al., 2018) and IGCV3 (Sun et al., 2018).

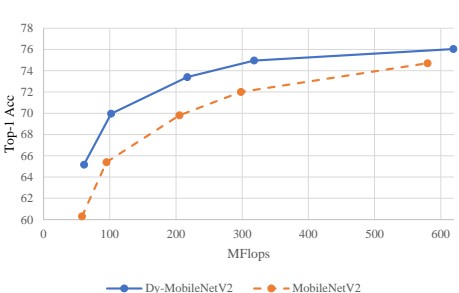

Figure 5: Compare with MobileNetV2 under the similar Flops constraint.

Table 1: Comparison of different network architectures over classification error and computation cost. The number in the brackets denotes the channel number controller (Sandler et al., 2018).

| Methods | MFLOPs | Top-1 err. (%) |
|---|---|---|
| Dy-shuffle (1.0) | 92 | 29.6 |
| Dy-mobile (1.0) | 135 | 28.27 |
| Dy-ResNet18 | 567 | 31.01 |
| Dy-ResNet50 | 1119 | 23.75 |
| ShuffleNet v1 (1.0) (Zhang et al., 2018) | 140 | 32.60 |
| MobileNet v2 (0.75) (Sandler et al., 2018) | 145 | 32.10 |
| MobileNet v2 (0.6) (Sandler et al., 2018) | 141 | 33.30 |
| MobileNet v1 (0.5)(Howard et al., 2017) | 149 | 36.30 |
| DenseNet (1.0) (Huang et al., 2017) | 142 | 45.20 |
| Xception (1.0) (Chollet, 2017) | 145 | 34.10 |
| IGCV2 (0.5) (Xie et al., 2018) | 156 | 34.50 |
| IGCV3-D (0.7) (Sun et al., 2018) | 210 | 31.50 |
| ShuffleNet V2 (1.0) (Ma et al., 2018) | 146 | 30.60 |
| MobileNetV2 (1.0) (Sandler et al., 2018) | 298 | 28.00 |
| ResNet18 | 1730 | 30.41 |
| ResNet50 | 3890 | 23.67 |

### 4.3 EXPERIMENT RESULTS AND ANALYSIS

**Analysis of accuracy and computation cost**   We demonstrate the results in Table 1, where the number in the brackets indicates the channel number controller (Sandler et al., 2018). We partitioned the result table into three parts: (1) The proposed dynamic networks; (2) Compared state-of-the-art networks under mobile settings; (3) The original networks corresponding to the implemented dynamic networks.

Table 1 provides several valuable observations: (1) Compared with these well known models under mobile setting, the proposed Dy-mobile and Dy-shuffle achieves the best classification error with lowest computation cost. This demonstrates that the proposed dynamic convolution is a simple yet effective way to reduce computation cost. (2) Compared with the corresponding basic neural structures, the proposed Dy-shuffle (1.0), Dy-mobile (1.0), Dy-ResNet18 and Dy-ResNet50 reduce 37.0%, 54.7%, 67.2% and 71.3% computation cost respectively with little drop on Top-1 accuracy. This shows that even though the proposed network significantly reduces the convolution computation cost, the generated dynamic kernel can still capture sufficient information from image contents. The results also indicate that the proposed dynamic convolution is a powerful plugin, which can be implemented on convolution layers to reduce computation cost while maintaining the accuracy.

Furthermore, we conduct detailed experiments on MobileNetV2. We replace the conventional convolution with the proposed dynamic one and get Dy-MobileNetV2. The accuracy of classification for models with different number of channels are shown in Figure 5. It is observed that Dy-MobileNetV2 consistently outperforms MobileNetV2 but the ascendancy is weaken with the increase of number of channels.

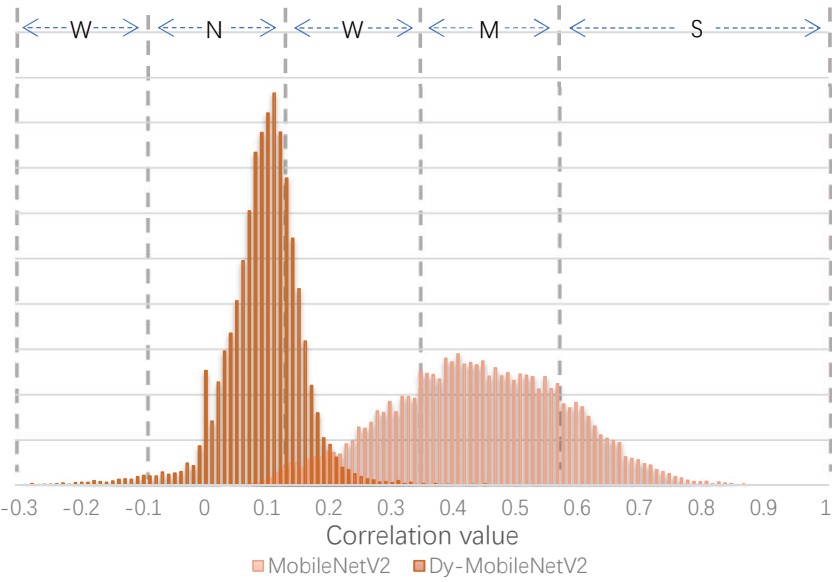

Figure 6: The correlation distribution of fixed kernel and the generated dynamic kernel, S, M, W, N denote strong, middle, weak and no correlation respectively. We can observe that compared with conventional fixed kernels, the generated dynamic kernels have small correlation values.

**Analysis of the dynamic kernel**   Aside from the quantitative analysis, we also demonstrate the redundancy of the generated dynamic kernels compared with conventional fixed kernels in Figure 6. We calculate the correlation between each feature maps output by the second last stage for the original MobileNetV2(1.0) and Dy-MobileNetV2 (1.0). Note that Dy-MobileNetV2 (1.0) is different with Dy-mobile(1.0). Dy-MobileNetV2(1.0) keeps the channels of each layer the same as the original one, while replace the conventional convolution with dynamic convolution. As shown in Figure 6, we can observe that the correlation distribution of dynamic kernels have more values distributed between −0.1 and 0.2 compared with fixed convolution kernel, which indicates that the redundancy between dynamic convolution kernels are much smaller than the fixed convolution kernels.

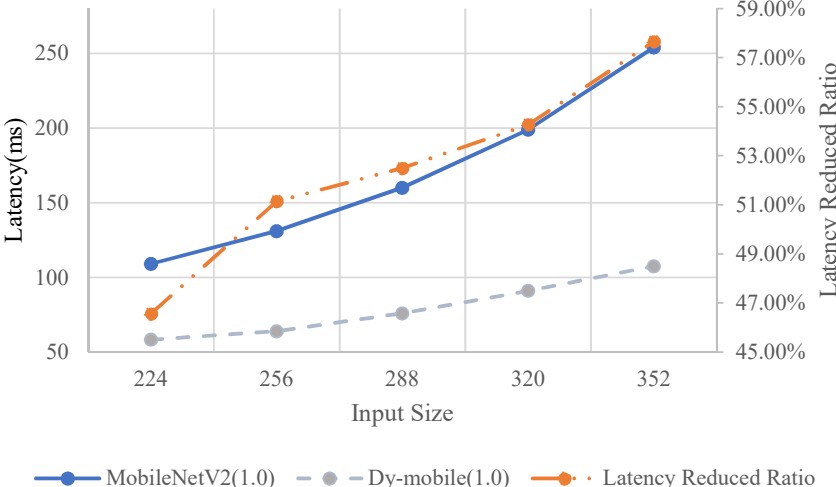

Figure 7: Latency for different input size.If we denote the latency of MobileNetV2(1.0),Dy-mobile as $L_{Fix}$ and $L_{Dym}$, then Latency Reduced Ratio is defined as $100\% - \frac{L_{Dym}}{L_{Fix}}$.

**Analysis of speed on the hardware**    We also analysis the inference speed of DyNet. We carry out experiments on the CPU platform (Intel(R) Core(TM) i7-7700 CPU @ 3.60GHz) with Caffe (Jia et al., 2014). We set the size of input as 224 and report the average inference time of 50 iterations. It is reasonable to set mini-batch size as 1, which is consistent with most inference scenarios. The results are shown in Table 2. Moreover, the latency of fusing fixed kernels is independent with the input size, thus we expect to achieve bigger acceleration ratio when the input size of networks become larger. We conduct experiments to verify this assumption, the results are shown in Figure 7. We can observe that the ratio of reduced latency achieved by DyNet gets bigger as the input size becomes larger. As shown in (Tan & Le, 2019), a larger input size can make networks perform significantly better, thus DyNet is more effective in this scenario.

We also analysis the training speed on the GPU platform. The model is trained with 32 NVIDIA Tesla V100 GPUs and the batch size is 2048. We report the average training time of one iteration in Table 2. It is observed that the training speed of DyNet is slower, it is reasonable because we fuse feature maps rather than kernels according to Eq. 2 in the training stage.

Table 2: Speed on the hardware.

| Methods | Top-1 err. (%) | Inference Time | Training Time |
|---|---|---|---|
| Dy-mobile(1.0) | 28.27 | 58.3ms | 250ms |
| MobileNetV2(1.0) | 28.00 | 109.1ms | 173ms |
| Dy-ResNet18 | 31.01 | 68.7ms | 213ms |
| ResNet18 | 30.41 | 90.7ms | 170ms |
| Dy-ResNet50 | 23.75 | 135.1ms | 510ms |
| ResNet50 | 23.67 | 199.6ms | 308ms |

## 4.4    EXPERIMENTS ON SEGMENTATION

To verify the scalability of DyNet on other tasks, we conduct experiments on segmentation. Compared to the method Dilated FCN with ResNet50 as basenet (Fu et al., 2018), Dilated FCN with Dy-ResNet50 reduces 69.3% FLOPs while maintaining the MIoU on Cityscapes validation set. The result are shown in Table 3.

Table 3: Experiments of segmentation on Cityscapes val set.

| Methods | BaseNet | GFLOPs | Mean IoU% |
|---|---|---|---|
| Dilated FCN(Fu et al., 2018) | ResNet50 | 310.8 | 70.03 |
| Dilated FCN(Fu et al., 2018) | Dy-ResNet50 | 95.6 | 70.48 |

## 4.5 ABLATION STUDY

**Comparison between dynamic convolution and static convolution** We correspondingly design two networks without dynamic convolution. Specifically, we remove the correlation prediction module and use fixed convolution kernel for Dy-mobile (1.0) and Dy-shuffle (1.5), and we keep the channel number the same as the dynamic convolution neural networks. We denote the baseline networks as Fix-mobile(1.0) and Fix-shuffle (1.5) respectively. The results are shown in Table 4, compare with baseline networks Fix-mobile (1.0) and Fix-shuffle (1.5), the proposed Dy-mobile (1.0) and Dy-shuffle (1.5) achieve absolute classification improvements by 5.19% and 2.82% respectively. This shows that directly decreasing the channel number to reduce computation cost influences the classification performance a lot. While the proposed dynamic kernel can retain the representation ability as mush as possible.

Table 4: Ablation experiments results of dynamic convolution and fixed convolution.

| Methods | MParams | MFLOPs | Top-1 err. (%) |
|---|---|---|---|
| Dy-mobile (1.0) | 7.36 | 135 | 28.27 |
| Dy-shuffle (1.5) | 11.0 | 180 | 27.48 |
| Fix-mobile (1.0) | 2.16 | 129 | 33.57 |
| Fix-shuffle (1.5) | 2.47 | 171 | 30.30 |

Table 5: Ablation experiments on $g_t$.

| Methods | MParams | MFLOPs | Top-1 err. (%) |
|---|---|---|---|
| Fix-mobile(1.0) | 2.16 | 129 | 33.57 |
| Dy-mobile(1.0, $g_t = 2$) | 3.58 | 131 | 29.43 |
| Dy-mobile(1.0, $g_t = 4$) | 5.47 | 133 | 28.69 |
| Dy-mobile(1.0, $g_t = 6$) | 7.36 | 135 | 28.27 |

**Effectiveness of $g_t$ for dynamic kernel** The group size $g_t$ in Eq. 1 does not change the computation cost of dynamic convolution, but affects the performance of network. Thus we provide ablative study on $g_t$. We set $g_t$ as 2,4,6 for dy-mobile(1.0) respectively and the results are shown in Table 5. The performance of dy-mobile(1.0) becomes better when $g_t$ gets larger. It is reasonable because larger $g_t$ means the number of kernels cooperated for obtaining one noise-irrelevant feature becomes larger. When $g_t = 1$, the coefficient prediction module can be regarded as merely learning the attention for different channels, which can improve the performance of networks as well (Hu et al., 2018). Therefore we provide ablative study for comparing $g_t = 1$ and $g_t = 6$ on Dy-mobile(1.0) and Dy-ResNet18. The results are shown in Table 6. From the table we can see that, setting $g_t = 1$ will reduce the Top-1 accuracy on ImageNet for Dy-mobile(1.0) and Dy-ResNet18 by 2.58% and 2.79% respectively. It proves that the improvement of our proposed dynamic networks does not only come from the attention mechanism.

Table 6: Comparison for $g_t = 1$ and $g_t = 6$.

| Methods | MParams | MFLOPs | Top-1 err. (%) |
|---|---|---|---|
| Dy-mobile (1.0, $g_t = 1$) | 2.64 | 131 | 30.85 |
| Dy-mobile (1.0, $g_t = 6$) | 7.36 | 135 | 28.27 |
| Dy-ResNet18 ($g_t = 1$) | 3.04 | 553 | 33.8 |
| Dy-ResNet18 ($g_t = 6$) | 16.6 | 567 | 31.01 |

## 5 CONCLUSION

In this paper, we propose a DyNet method to adaptively generate convolution kernels based on image content, which reduces the redundant computation cost existed in conventional fixed convolution kernels. Based on the proposed DyNet, we design several dynamic convolution neural networks based on well known architectures, i.e., Dy-mobile, Dy-shuffle, Dy-ResNet18, Dy-ResNet50. The experiment results show that DyNet reduces $37.0\%$, $54.7\%$, $67.2\%$ and $71.3\%$ FLOPs respectively, while maintaining the performance unchanged. As future work, we want to further explore the redundancy phenomenon existed in convolution kernels, and find other ways to reduce computation cost, such as dynamically aggregate different kernels for different images other than fixed groups used in this paper.

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

# A    APPENDIX

## A.1    DETAILED ANALYSIS OF OUR MOTIVATION

We illustrate our motivation from a convolution with output $f(x)$, i.e.,

$$f(x) = x \otimes w, \tag{4}$$

where $\otimes$ denotes the convolutional operator, $x \in R^n$ is a vectorized input and $w \in R^n$ means the filter. Specifically, the $i_{th}$ element of the convolution output $f(x)$ is calculated as:

$$f_i(x) = \langle x_{(i)}, w \rangle, \tag{5}$$

where $\langle \cdot, \cdot \rangle$ provides an inner product and $x_{(i)}$ is the circular shift of $x$ by $i$ elements. We define the index $i$ started from 0.

We denote the noises in $x_{(i)}$ as $\sum_{j=0}^{d-1} \alpha_j y_j$, where $\alpha_j \in R$ and $\{y_0, y_1, ..., y_{d-1}\}$ are the base vectors of noise space $\Psi$. Then the kernels in one convolutional layer can be represented as $\{w_0, w_1, ..., w_c\}$. The space expanded by $\{w_0, w_1, ..., w_c\}$ is $\Omega$. We can prove if the kernels are trained until $\Psi \subset \Omega$, then for each $w_k \notin \Psi$, we can get the noise-irrelevant $f_i(x^{white}) = \langle x_{(i)}^{white}, w_k \rangle$ by the cooperation of other kernels $w_0, w_1, ...$.

Firstly $x_{(i)}$ can be decomposed as:

$$x_{(i)} = \bar{x}_{(i)} + \beta w_k + \sum_{j=0}^{d-1} \alpha_j y_j, \tag{6}$$

where $\beta \in R$ and $\bar{x} \in R^n$ is vertical to $w_k$ and $y_j$.

For concision we assume the norm of $w_k$ and $y_j$ is 1. Then,

$$f_i(x) = \langle x_{(i)}, w_k \rangle = \langle \bar{x}_{(i)} + \beta w_k + \sum_{j=0}^{d-1} \alpha_j y_j, w_k \rangle = \beta \langle w_k, w_k \rangle + \sum_{j=0}^{d-1} \alpha_j \langle y_j, w_k \rangle \tag{7}$$

When there is no noise, i.e. $\alpha_j = 0$ for $j = 0, 1, ..., d - 1$, the white output $f_i(x^{white})$ becomes:

$$f_i(x^{white}) = \langle x_{(i)}^{white}, w_k \rangle = \langle \bar{x}_{(i)} + \beta w_k, w_k \rangle = \beta \langle w_k, w_k \rangle = \beta. \tag{8}$$

It is proved in the Appendix A.2 that:

$$f_i(x^{white}) = \langle a_{00} w_k + \sum_t \beta_t w_t, x_{(i)} \rangle = (a_{00} + \beta_k) \langle w_k, x_{(i)} \rangle + \sum_{t \neq k} \beta_t \langle w_t, x_{(i)} \rangle, \tag{9}$$

where $\beta_0, ..., \beta_c$ is determined by the input image.

Eq. 9 is fulfilled by linearly combine convolution output $\langle w_k, x_{(i)} \rangle$ and $\langle w_t, x_{(i)} \rangle$ for those $\beta_t \neq 0$ in the following layers. Thus if there are $N$ coefficients in Eq. 9 that are not 0, then we need to carry out $N$ times convolution operation to get the noise-irrelevant output of kernel $w_t$, this causes redundant calculation.

In Eq. 9, we can observe that the computation cost can be reduced to one convolution operation by linearly fusing those kernels to a dynamic one:

$$\widetilde{w} = (a_{00} + \beta_k) w_k + \sum_{t \neq k, \beta_t \neq 0} \beta_t w_t$$
$$f_i(x^{white}) = \langle \widetilde{w}, x_{(i)} \rangle. \tag{10}$$

In Eq. 10, the coefficients $\beta_0, \beta_1, ...$ is determined by $\alpha_0, \alpha_1, ...,$ thus they should be generated based on the input of network. *This is the motivation of our proposed dynamic convolution.*

## A.2    PROVING OF EQ. 9

We denote $g_{ij}(x)$ as $\langle x_{(i)}, y_j \rangle$, $j = 0, 1, ..., d - 1$. Then,

$$g_{ij}(x) = \langle x_{(i)}, y_j \rangle = \langle \bar{x}_{(i)} + \beta w_k + \sum_{t=0}^{d-1} \alpha_t y_t, y_j \rangle = \beta \langle w_k, y_j \rangle + \sum_{t=0}^{d-1} \alpha_t \langle y_t, yj \rangle. \tag{11}$$

By summarize Eq. 7 and Eq. 11, we get the following equation:

$$
\begin{bmatrix}
\langle w_k, w_k \rangle & \langle y_0, w_k \rangle & \langle y_1, w_k \rangle & \cdots & \langle y_{d-1}, w_k \rangle \\
\langle w_k, y_0 \rangle & \langle y_0, y_0 \rangle & \langle y_1, y_0 \rangle & \cdots & \langle y_{d-1}, y_0 \rangle \\
\langle w_k, y_1 \rangle & \langle y_0, y_1 \rangle & \langle y_1, y_1 \rangle & \cdots & \langle y_{d-1}, y_1 \rangle \\
\vdots & \vdots & \vdots & \cdots & \vdots \\
\langle w_k, y_{d-1} \rangle & \langle y_0, y_{d-1} \rangle & \cdots & \cdots & \langle y_{d-1}, y_{d-1} \rangle
\end{bmatrix}
\begin{bmatrix}
\beta \\ \alpha_0 \\ \alpha_1 \\ \vdots \\ \alpha_{d-1}
\end{bmatrix}
=
\begin{bmatrix}
f_i(x) \\ g_{i0}(x) \\ g_{i1}(x) \\ \vdots \\ g_{i(d-1)}(x)
\end{bmatrix},
\tag{12}
$$

We simplify this equation as:

$$
A\vec{x} = \vec{b}.
\tag{13}
$$

Because $w_k \notin \Psi$, we can denote $w_k$ as:

$$
w_k = \gamma_\perp w_\perp + \sum_{j=0}^{d-1} \gamma_j y_j,
\tag{14}
$$

where $w_\perp$ is vertical to $y_0, ..., y_{d-1}$ and $\gamma_\perp \neq 0$.
moreover because $|w_k| = 1$, thus

$$
|\gamma_\perp|^2 + \sum_{j=0}^{d-1} |\gamma_j|^2 = 1.
\tag{15}
$$

It can be easily proved that:

$$
A =
\begin{bmatrix}
1 & \gamma_0 & \gamma_1 & \cdots & \gamma_{d-1} \\
\gamma_0 & 1 & 0 & \cdots & 0 \\
\gamma_1 & 0 & 1 & \cdots & 0 \\
\vdots & \vdots & \vdots & \cdots & \vdots \\
\gamma_{d-1} & 0 & \cdots & \cdots & 1
\end{bmatrix}.
\tag{16}
$$

thus,

$$
\begin{aligned}
|A| &=
\begin{vmatrix}
1 & \gamma_0 & \gamma_1 & \cdots & \gamma_{d-1} \\
\gamma_0 & 1 & 0 & \cdots & 0 \\
\gamma_1 & 0 & 1 & \cdots & 0 \\
\vdots & \vdots & \vdots & \cdots & \vdots \\
\gamma_{d-1} & 0 & \cdots & \cdots & 1
\end{vmatrix} \\[2mm]
&=
\begin{vmatrix}
1 - \gamma_0^2 & 0 & \gamma_1 & \cdots & \gamma_{d-1} \\
\gamma_0 & 1 & 0 & \cdots & 0 \\
\gamma_1 & 0 & 1 & \cdots & 0 \\
\vdots & \vdots & \vdots & \cdots & \vdots \\
\gamma_{d-1} & 0 & \cdots & \cdots & 1
\end{vmatrix} \\[2mm]
&=
\begin{vmatrix}
1 - \gamma_0^2 - \gamma_1^2 & 0 & 0 & \cdots & \gamma_{d-1} \\
\gamma_0 & 1 & 0 & \cdots & 0 \\
\gamma_1 & 0 & 1 & \cdots & 0 \\
\vdots & & \vdots & \vdots & \cdots & \vdots \\
\gamma_{d-1} & & 0 & \cdots & \cdots & 1
\end{vmatrix} \\[2mm]
&=
\begin{vmatrix}
1 - \gamma_0^2 - \gamma_1^2 - \cdots - \gamma_{d-1}^2 & 0 & 0 & \cdots & 0 \\
\gamma_0 & 1 & 0 & \cdots & 0 \\
\gamma_1 & 0 & 1 & \cdots & 0 \\
\vdots & & \vdots & \vdots & \cdots & \vdots \\
\gamma_{d-1} & & 0 & \cdots & \cdots & 1
\end{vmatrix} \\[2mm]
&=
\begin{vmatrix}
\gamma_\perp^2 & 0 & 0 & \cdots & 0 \\
\gamma_0 & 1 & 0 & \cdots & 0 \\
\gamma_1 & 0 & 1 & \cdots & 0 \\
\vdots & \vdots & \vdots & \cdots & \vdots \\
\gamma_{d-1} & 0 & \cdots & \cdots & 1
\end{vmatrix} \\[2mm]
&= \gamma_\perp^2 \neq 0.
\end{aligned}
\tag{17}
$$

thus,

$$\vec{x} = A^{-1}\vec{b}. \tag{18}$$

If we denote the elements of the first row of $A^{-1}$ as $a_{00}, a_{01}, ..., a_{0d}$, then

$$f_i(x^{white}) = \beta = a_{00}f_i(x) + \sum_{j=0}^{d-1} a_{0(j+1)}g_{i,j}(x)$$

$$= a_{00}\langle w_k, x_{(i)}\rangle + \sum_{j=0}^{d-1} a_{0(j+1)}\langle y_j, x_{(i)}\rangle \tag{19}$$

$$= \langle a_{00}w_k + \sum_{j=0}^{d-1} a_{0(j+1)}y_j, x_{(i)}\rangle.$$

Because $\Psi \subset \Omega$, there exists $\{\beta_t \in R | t = 0, 1, ..., c\}$ that

$$\sum_{j=0}^{d-1} a_{0(j+1)}y_j = \sum_t \beta_t w_t. \tag{20}$$

Then,

$$f_i(x^{white}) = \langle a_{00}w_k + \sum_t \beta_t w_t, x_{(i)}\rangle = (a_{00} + \beta_k)\langle w_k, x_{(i)}\rangle + \sum_{t\neq k} \beta_t \langle w_t, x_{(i)}\rangle, \tag{21}$$

