# OpenReview forum: "DyNet: Dynamic Convolution for Accelerating Convolution Neural Networks"
_ICLR.cc/2020/Conference — Reject_

### Official Review · AnonReviewer2 · 2019-10-23
**Official Blind Review #1718**

**Rating:** 3

**Review:**


Main contribution of the paper
- The paper proposes a dynamic convolution selection method can be applied to arbitrary classification networks based on the global average pooled (GAP) feature map info.
- The method obtained improvements over various networks (SuffleNet v2, MobileNet v2, ResNet 18) on ImageNet.

Methods
- Given the set of fixed convolutional filters, the method dynamically selects the (weighted sum) kernels by given a kind of channel attention.
- The GAP of the features gives the channel attention on each stage, and the method applies the dynamic selection of the kernels.
- The number of channels in skip-connection shluld be the same because it should be elementwise multiplied with the channel attention acquired from GAP.
- The author slightly revises the baseline networks to set the networks integrated with the proposed method to have smaller Flops.

Questions
- According to Figure 4, it seems that the proposed add-on requires many parameters because it would include a FC layer for each block. But we cannot find the number of parameters in this paper.
- The parameter $g_t$ is defined as 6. The experiment shows the ablation to the case $g_t$ =1, but what if we set the parameters to other numbers?

Strong points
- The proposed model achieved improvement with fewer Flops on large scale image classification dataset.
- The method shows effectiveness when it is attached to various classification networks.

Concerns
- The main concern of the reviewer is that the model shares the core contribution to the existing method; squeeze-and-excitation network (SEnet, Hu et.al.). The method also proposes the attention-based scaling of channels, where the attention comes from GAP, so the reviewer thinks that it is possible to explain this work as some variation of SEnet.
The author should clarify the difference and the strong points of the proposed block compared to SEnet.
- Also, the reviewer cannot guarantee that the networks trained by the proposed method can transfer the knowledge to other tasks such as detection.
The reviewer thinks that it is a critical part because one of the primal reasons for training the network is to use them as the pre-trained backbone for the other tasks.
Regarding this, the baseline methods (MobileNet V2, Shufflenet v2, ResNet)  are widely used as a pre-trained backbone for object detection, and the papers mention the CoCo object detection results using the pre-trained backbones from their method. The reviewer thinks that the experiment regarding this should be included.
- The other thing is that the parameter increases. As in the question, the reviewer thinks that the number of parameters would be increased. The reviewer agrees that some recent works focus more on Flops, but the number of parameters is also discussed in general, when telling about the 'model size'.

Conclusion
- The author proposed a dynamic kernel selection method (add-on), which can enhance the classification accuracy of the baseline network.
- However, the reviewer cannot convince the novelty of the proposed approach and usefulness of the pre-trained backbone network from the proposed method when applying it to the other tasks (Object detection).


Inquiries
- Clarifying the difference between SEnet.
- Testing the ImageNet trained network of the proposed method into an object detection task (as the pre-trained backbone).
- Discussing the number of the parameter as well.


**Experience Assessment:**

I have read many papers in this area.

**Review Assessment: Checking Correctness Of Derivations And Theory:**

I assessed the sensibility of the derivations and theory.

**Review Assessment: Checking Correctness Of Experiments:**

I carefully checked the experiments.

**Review Assessment: Thoroughness In Paper Reading:**

I read the paper thoroughly.

---

> ### Author Response · Authors · 2019-11-11
> **Thanks for the detailed response. Please see comments below.**
>
> >>> Response to “The proposed add-on requires many parameters but the number of parameters is not shown in this paper”:
> Thanks for your suggestion, we will add the number of parameters in experiment tables in the revision.
> >>> Response to “The experiment shows the ablation to the case g_t=1, but what if we set the parameters to other numbers?”:
> Thanks for your comments! Actually, we have shown it in Table 5. The value of g_t does not change the computation cost of convolution but affects the performance of networks. The performance will become better when g_t gets larger.
> >>> Response to “The author should clarify the difference and the strong points of the proposed block compared to SEnet.”
> Thanks for your meaningful question!
> From the method's point of view, we fuse feature maps instead of fusing kernels in the training stage, it can be done via the combination of group point-wise convolution and SE mechanism. However we fuse kernels in the inference stage, it can`t be done via this combination. This is the core difference with SEnet and reduces the computation cost by gt times compared with fusing feature maps or the combination of group point-wise convolution and SE mechanism.
> From the experiments' point of view, as discussed in Sec 4.5, our proposed method will be the same as SENet when g_t=1. However, compared with g_t=6, its performance drops a lot while the computation cost of convolution stay the same as shown in Table 6.
> >>> Response to “Testing the ImageNet trained network of the proposed method into an object detection task (as the pre-trained backbone).”
> Thanks for your suggestion. We are trying to implement the baseline and add experiments for detection in the revision before the deadline.

---

> > ### Author Response · Authors · 2019-11-15
> > **The revision is submitted and the points you concern have been updated.**
> >
> > >>> Response to “The proposed add-on requires many parameters but the number of parameters is not shown in this paper”:
> > The proposed add-on indeed requires many parameters, we have added the number of parameters in Table4, Table5 and Table6 to illustrate this point.
> >                               Table 4
> > +------------------+---------+--------+----------------+
> > | Methods          | MParams | MFLOPs | Top-1 err. (%) |
> > +------------------+---------+--------+----------------+
> > | Dy-mobile(1.0)   | 7.36    | 135    | 28.38          |
> > +------------------+---------+--------+----------------+
> > | Dy-shuffle(1.5)  | 11      | 180    | 27.48          |
> > +------------------+---------+--------+----------------+
> > | Fix-mobile(1.0)  | 2.16    | 129    | 33.57          |
> > +------------------+---------+--------+----------------+
> > | Fix-shuffle(1.5) | 2.47    | 171    | 30.3           |
> > +------------------+---------+--------+----------------+
> >                                 Table 5
> > +-----------------+---------+--------+----------------+
> > | Methods         | MParams | MFLOPs | Top-1 err. (%) |
> > +-----------------+---------+--------+----------------+
> > | Fix-mobile(1.0) | 2.16    | 129    | 33.57          |
> > +-----------------+---------+--------+----------------+
> > | Dy-mobile(gt=2) | 3.58    | 131    | 29.43          |
> > +-----------------+---------+--------+----------------+
> > | Dy-mobile(gt=4) | 5.47    | 133    | 28.69          |
> > +-----------------+---------+--------+----------------+
> > | Dy-mobile(gt=6) | 7.36    | 135    | 28.38          |
> > +-----------------+---------+--------+----------------+
> >                              Table 6
> > +---------------------+---------+--------+----------------+
> > | Methods             | MParams | MFLOPs | Top-1 err. (%) |
> > +---------------------+---------+--------+----------------+
> > | Dy-mobile(1.0,gt=1) | 2.64    | 131    | 30.85          |
> > +---------------------+---------+--------+----------------+
> > | Dy-mobile(1.0,gt=6) | 7.36    | 135    | 28.27          |
> > +---------------------+---------+--------+----------------+
> > | Dy-ResNet18(gt=1)   | 3.04    | 556    | 33.8           |
> > +---------------------+---------+--------+----------------+
> > | Dy-ResNet18(gt=6)   | 16.6    | 567    | 31.01          |
> > +---------------------+---------+--------+----------------+

---

### Official Review · AnonReviewer1 · 2019-10-23
**Official Blind Review #1**

**Rating:** 6

**Review:**

=== Summary ===
The authors propose to use dynamic convolutional kernels as a means to reduce the computation cost in static CNNs while maintaining their performance. The dynamic kernels are obtained by a linear combination of static kernels where the weights of the linear combination are input-dependent (they are obtained similarly to the coefficients in squeeze-and-excite). The authors also include a theoretical and experimental study of the correlation.
The authors conduct extensive experiments on image classification and segmentation and show that dynamic convolutional kernels with reduced number of channels lead to significant reduction in FLOPS and increase in inference speed (for batch size 1) compared to their static counterparts with higher number of channels.

=== Recommendation ===
The experimental setup is rigorous but the current draft lacks some metrics that should be reported (as training times, parameter counts, memory requirements at training/inference) since the focus is on making CNNs more efficient.

The presented experimental results are satisfactory but the studied networks are not quite SOTA: they are much more competitive alternatives to ResNet and MobileNetv2. The correlation study is interesting.

My main issue with the paper is the lack of novelty. The use of dynamic convolutions is by no means a novel idea and has been studied in multiple previous works in vision (mixture of experts, soft conditional computation, pay less attention with dynamic convolutions, ...) which the authors fail to cite/compare against.
However, most previous work focuses on leveraging dynamic kernels to use more parameters so the focus on accelerating CNNs is novel.

Overall, I am on the fence with this paper but slightly leaning towards rejecting it for the above reasons.

=== Questions/Comments ===
- Figure 5: how are the models constrained to have same FLOPS? Is it by changing the number of channels?
- Consider adding training times for more transparency
- Consider adding parameter counts in experiment tables
- The related work subsection 2.3 is rather poor compared to existing work.
- 'While model compression based methods' -> 'On the other hand, model compression based methods'
- 'computing efficient' -> 'compute efficient'
- 'values distribute' -> 'values distributed'
- 'DETAIL ANALYSIS OF OUR MOTIVATION' -> 'Detailed analysis of our motivation'

**Experience Assessment:**

I have published one or two papers in this area.

**Review Assessment: Checking Correctness Of Derivations And Theory:**

I assessed the sensibility of the derivations and theory.

**Review Assessment: Checking Correctness Of Experiments:**

I carefully checked the experiments.

**Review Assessment: Thoroughness In Paper Reading:**

I read the paper thoroughly.

---

> ### Author Response · Authors · 2019-11-11
> **Thanks for the detailed response. Please see comments below.**
>
> >>> Response to “Figure 5: how are the models constrained to have same FLOPS?”:
> Thanks for your valuable question which help us realize some unclear statements. By changing the number of channels of MobileNetV2, we can get MobileNetV2(0.35), MobileNetV2(0.5), MobileNetV2(0.75), MobileNetV2(1.0) and MobileNetV2(1.4). To constrain our models to have the same FLOPS with them, we keep the channels of each layer the same as the original one, while replacing the conventional convolution with dynamic convolution. If we ignore the additional FLOPs of coefficient prediction module and dynamic generation module, which is negligible, the FLOPs will stay the same. This result shows that our proposed dynamic convolution can be deployed as a plug-and-play unit to replace conventional convolution.
> We realize that it is not proper to call these models “Dy-mobile” because they have different structures. In the updated version we will call them as Dy-MobileNetV2(0.35), Dy-MobileNetV2(0.5), Dy-MobileNetV2(0.75), Dy-MobileNetV2(1.0) and Dy-MobileNetV2(1.4).
> >>> Response to “Consider adding training times and parameter counts.”:
> Thanks for your suggestion! We will add parameter counts in experiment tables and discuss the training time in Sec 4.3 in the revision.
> >>> Response to “Lack of novelty.”:
> For previous works on dynamic convolution, they all directly generate convolution kernels via a linear layer (including “pay less attention with dynamic convolutions”). In computer vision tasks, the parameter counts of convolution is large, which makes the number of parameters of the linear layer unbearable. In our proposed method, the linear layer is merely used to predict the coefficients for linearly combining static kernels. It can solve this problem and thus be used to achieve real speed up for CNN on hardware.
> The most related work is “soft conditional computation”. They focus on using more parameters to make models to be more expressive while we focus on reducing redundant calculations in convolution. According to the theoretical analysis in appendix A and correlation study in Figure 6, we find that correlations among convolutional kernels can be reduced via dynamically fusing several kernels. Thus different from “soft conditional computation” which replaces the conventional convolution directly, we recommend to reduce the channel numbers (we reduce half of the channels for convolution layers in Dy-ResNet18/50 and Dy-shuffle) and then replace conventional convolution with dynamic one. We think this may bring a larger improvement. For example, compared with directly replace the conventional convolution in ResNet18, Dy-ResNet50 reduce 37.9% FLOPs while improves 2.03% top-1 accuracy on ImageNet. Moreover, since this paper is submitted to NeurIPS2019 this May as well, it may be regarded as a concurrent work
> We will add a comparison against existing work in Sec 2.3.

---

> > ### Comment · AnonReviewer1 · 2019-11-14
> > **Reviewer reply**
> >
> > I read the authors' replies to the reviews and changed my score to weak accept.
> > I'm still on the fence as there is no comparison to previous work.
> >
> > I think this could be a better paper if the authors take into account the reviewer's suggestions. Most importantly, the authors should simply state all metrics (inference/training latency, parameters, FLOPS (including the coefficient prediction module) for the benchmarked networks and clearly state the dynamic convolutions have been proposed before.
> >
> > Note that pay less attention with dynamic convolutions predicts the parameters of a depthwise convolution with some additional weight sharing across layers which reduces the "makes the number of parameters of the linear layer unbearable" issue.

---

> > > ### Author Response · Authors · 2019-11-15
> > > **Thanks for your helpful reply and we have updated our paper.**
> > >
> > > >>> Response to "the authors should simply state all metrics (inference/training latency, parameters, FLOPS (including the coefficient prediction module) for the benchmarked networks":
> > >
> > > Thanks for your suggestion. We have updated the experiment tables to show the training latency and amount of parameters. The FLOPs of coefficient prediction module and dynamic generation module is taken into account in this version (Figure5, Table1, Table2, Table4, Table5, Table6).  Because it has been shown in the Table2-6 that the proposed method increases the number of parameters and training latency, we don't add the parameter amounts in Table 1 in order to highlight the advantage is reducing computation cost while maintaining the accuracy.
> > >
> > > In Table 2, we add the training latency:
> > >                                 Table2
> > > +------------------+------------------+-----------------+----------------+
> > > | Method           | Top-1 error. (%) | Inference Time  | Training Time  |
> > > +------------------+------------------+-----------------+----------------+
> > > | Dy-mobile(1.0)   | 28.27            | 58.3ms          | 250ms          |
> > > +------------------+------------------+-----------------+----------------+
> > > | MobileNetV2(1.0) | 28               | 109.1ms         | 173ms          |
> > > +------------------+------------------+-----------------+----------------+
> > > | Dy-ResNet18      | 31.01            | 68.7ms          | 213ms          |
> > > +------------------+------------------+-----------------+----------------+
> > > | ResNet18         | 30.41            | 90.7ms          | 170ms          |
> > > +------------------+------------------+-----------------+----------------+
> > > | Dy-ResNet50      | 23.75            | 135.146ms       | 510ms          |
> > > +------------------+------------------+-----------------+----------------+
> > > | ResNet50         | 23.67            | 199.6ms         | 308ms          |
> > > +------------------+------------------+-----------------+----------------+
> > >
> > >
> > > In table 4, table 5 and table 6, we add the amount of parameters and and take the FLOPs of coefficient prediction module and dynamic generation module into account:
> > >
> > >                               Table 4
> > > +------------------+---------+--------+----------------+
> > > | Methods          | MParams | MFLOPs | Top-1 err. (%) |
> > > +------------------+---------+--------+----------------+
> > > | Dy-mobile(1.0)   | 7.36    | 135    | 28.38          |
> > > +------------------+---------+--------+----------------+
> > > | Dy-shuffle(1.5)  | 11      | 180    | 27.48          |
> > > +------------------+---------+--------+----------------+
> > > | Fix-mobile(1.0)  | 2.16    | 129    | 33.57          |
> > > +------------------+---------+--------+----------------+
> > > | Fix-shuffle(1.5) | 2.47    | 171    | 30.3           |
> > > +------------------+---------+--------+----------------+
> > >                                 Table 5
> > > +-----------------+---------+--------+----------------+
> > > | Methods         | MParams | MFLOPs | Top-1 err. (%) |
> > > +-----------------+---------+--------+----------------+
> > > | Fix-mobile(1.0) | 2.16    | 129    | 33.57          |
> > > +-----------------+---------+--------+----------------+
> > > | Dy-mobile(gt=2) | 3.58    | 131    | 29.43          |
> > > +-----------------+---------+--------+----------------+
> > > | Dy-mobile(gt=4) | 5.47    | 133    | 28.69          |
> > > +-----------------+---------+--------+----------------+
> > > | Dy-mobile(gt=6) | 7.36    | 135    | 28.38          |
> > > +-----------------+---------+--------+----------------+
> > >                              Table 6
> > > +---------------------+---------+--------+----------------+
> > > | Methods             | MParams | MFLOPs | Top-1 err. (%) |
> > > +---------------------+---------+--------+----------------+
> > > | Dy-mobile(1.0,gt=1) | 2.64    | 131    | 30.85          |
> > > +---------------------+---------+--------+----------------+
> > > | Dy-mobile(1.0,gt=6) | 7.36    | 135    | 28.27          |
> > > +---------------------+---------+--------+----------------+
> > > | Dy-ResNet18(gt=1)   | 3.04    | 556    | 33.8           |
> > > +---------------------+---------+--------+----------------+
> > > | Dy-ResNet18(gt=6)   | 16.6    | 567    | 31.01          |
> > > +---------------------+---------+--------+----------------+
> > >
> > > >>> Response to "the authors should clearly state the dynamic convolutions have been proposed before.”
> > > We have updated Sec 2.3 to compare with existing work and state that the idea of linearly combining static kernels using predicted coefficients has been proposed before.
> > >
> > > >>>Response to “Figure 5: how are the models constrained to have same FLOPS?” :
> > > We have updated Sec 4.3 and Figure 5 to illustrate this issue as follows:
> > > “Furthermore, we conduct detailed experiments on MobileNetV2. We replace the conventional convolution with the proposed dynamic one and get Dy-MobileNetV2. The accuracy of classification for models with different number of channels are shown in Figure 5. It is observed that Dy-MobileNetV2 consistently outperforms MobileNetV2 but the ascendancy is weaken with the increase of number of channels.”

---

### Official Review · AnonReviewer3 · 2019-10-24
**Official Blind Review #3**

**Rating:** 3

**Review:**

This paper proposed dynamic convolution (DyNet) to accelerating convolution networks. The new method is tested on the ImageNet dataset with three different backbones. It reduces the computation flops by a large margin while keeps similar classification accuracy. The additional segmentation experiment on the Cityscapes dataset also shows the new module can save computation a lot while maintaining similar segmentation accuracy.

Clarity:
The novelty of the paper is limited and the experimental results are weird for me.
1. The proposed module named dynamic convolution is detailed in Sec 3.2. As far as I can see, it is very similar to the former SENet especially in Figure (3) and Equation (2). The only difference is the introduction of g_t where the output dimension is much larger than SENet.

2. As shown in Equation (2), the proposed method contains the normal computation of fixed kernels. How can this method save computations compared to classical convolution? Is the computation flops calculated in the right way?

3.  The results in Table 5 are strange to me. Larger g_t will increase the flops absolutely according to Equation (2).

4. The author may need to show the comparisons of the number of parameters. In my opinion, the new module will increase the parameters a lot (the output dimension of the fully connected layer is as large as C_cout*g_t).


**Experience Assessment:**

I have published in this field for several years.

**Review Assessment: Checking Correctness Of Derivations And Theory:**

I assessed the sensibility of the derivations and theory.

**Review Assessment: Checking Correctness Of Experiments:**

I carefully checked the experiments.

**Review Assessment: Thoroughness In Paper Reading:**

I read the paper at least twice and used my best judgement in assessing the paper.

---

> ### Author Response · Authors · 2019-11-09
> **Thanks for the detailed response. Please see comments below.**
>
> >>> Response to “The only difference with SENet is the introduction of g_t where the output dimension is much larger than SENet.”:
> Thanks for your meaningful question! The motivation of our proposed approach is different from SENet. We focus on reducing the computation cost during inference stage, while SENet learns channel-wise importance to recalibrates feature maps. From the methods point of view, during training, indeed the only difference is that the output dimension is much larger than SENet. However, during inference, SENet recalibrate feature maps by learned weights, while we fuse kernels instead of fusing feature maps, which can reduce the output dimension and computation cost of convolution kernels.
> Moreover, as discussed in Sec 4.5, our proposed method will be the same as SENet when g_t=1. However, compared with g_t=6, its performance drops a lot while the computation cost of convolution stay the same as shown in Table 6.
> >>> Response to “How can this method save computations compared to classical convolution?”:
> Thanks for your valuable comments! The goal of our proposed approach is to reduce the computation cost in the inference stage. Equation (2) only shows that fusing feature maps is mathematically equivalent with fusing kernels. During training, we firstly generate C_cout*g_t output feature maps and then get C_out feature maps by fusing each g_t ones. Thus we can not save computation cost in training procedure. However, we fuse kernels during inference stage instead of fusing feature maps. After fusing, the weight for one convolution layer with shape [C_out*g_t, C_in, k, k] will become [C_cout, C_in, k, k], thus the FLOPs is reduced by g_t times.
> >>> Response to “Why larger g_t doesn`'t increase the flops in Table 5?”:
> Compared with g_t=1, the computation cost of convolution is 6 times larger when g_t=6 during training. However, we will fuse each g_t kernels to get dynamic convolutional weight with shape [C_cout, C_in, k, k] during inference as Equation (1), it makes the FLOPs of convolution invariable with g_t. As for the additional FLOPs of fusing kernels, it varies with g_t but is negligible. For Dy-mobile(1.0), the FLOPs of fusing kernels is merely 0.459*g_t M, when g_t=6 it is 2.75M while the FLOPs of convolution is 129 M.
> >>> Response to “The author may need to show the comparisons of the number of parameters.”:
> Thanks for the suggestion! The new module indeed increases the parameters a lot, because we mainly focus on the computation cost rather than the number of parameters. We will add parameter counts in experiment table in our updated version.

---

> > ### Author Response · Authors · 2019-11-15
> > **The revision is submitted and the points you concern have been updated.**
> >
> > >>> Response to “Why larger g_t doesn't increase the flops in Table 5?”
> > 	The FLOPs of coefficient prediction module and dynamic generation module is taken into account in this version (Figure5, Table1, Table2, Table4, Table5, Table6). Thus the flops in Table 5 increases slightly as g_t becoming larger.
> >                                 Table 5
> > +-----------------+---------+--------+----------------+
> > | Methods         | MParams | MFLOPs | Top-1 err. (%) |
> > +-----------------+---------+--------+----------------+
> > | Fix-mobile(1.0) | 2.16    | 129    | 33.57          |
> > +-----------------+---------+--------+----------------+
> > | Dy-mobile(gt=2) | 3.58    | 131    | 29.43          |
> > +-----------------+---------+--------+----------------+
> > | Dy-mobile(gt=4) | 5.47    | 133    | 28.69          |
> > +-----------------+---------+--------+----------------+
> > | Dy-mobile(gt=6) | 7.36    | 135    | 28.38          |
> > +-----------------+---------+--------+----------------+
> > >>> Response to “The author may need to show the comparisons of the number of parameters.”:
> > 	We have added the number of parameters in Table4, Table5 and Table6.
> >                               Table 4
> > +------------------+---------+--------+----------------+
> > | Methods          | MParams | MFLOPs | Top-1 err. (%) |
> > +------------------+---------+--------+----------------+
> > | Dy-mobile(1.0)   | 7.36    | 135    | 28.38          |
> > +------------------+---------+--------+----------------+
> > | Dy-shuffle(1.5)  | 11      | 180    | 27.48          |
> > +------------------+---------+--------+----------------+
> > | Fix-mobile(1.0)  | 2.16    | 129    | 33.57          |
> > +------------------+---------+--------+----------------+
> > | Fix-shuffle(1.5) | 2.47    | 171    | 30.3           |
> >
> >                              Table 6
> > +---------------------+---------+--------+----------------+
> > | Methods             | MParams | MFLOPs | Top-1 err. (%) |
> > +---------------------+---------+--------+----------------+
> > | Dy-mobile(1.0,gt=1) | 2.64    | 131    | 30.85          |
> > +---------------------+---------+--------+----------------+
> > | Dy-mobile(1.0,gt=6) | 7.36    | 135    | 28.27          |
> > +---------------------+---------+--------+----------------+
> > | Dy-ResNet18(gt=1)   | 3.04    | 556    | 33.8           |
> > +---------------------+---------+--------+----------------+
> > | Dy-ResNet18(gt=6)   | 16.6    | 567    | 31.01          |
> > +---------------------+---------+--------+----------------+

---

### Public Comment · ~Brandon_Yang1 · 2019-10-23
**Related work**

Hi,

I found this to be a very interesting analysis on using input-dependent convolutional kernels. It might be worth noting that our previous work also studies the benefits of using input-dependent convolutional kernels over static convolutional kernels in the image classification and detection settings: https://arxiv.org/abs/1904.04971 . I’d be very interested in better understanding the differences between our two approaches.

I also had a clarifying question about the work. In Table 5, why does increasing the group size (g_t) not change the MFLOPs of the network? In my understanding, the coefficient prediction module and dynamic generation module would both require more computation with larger values of g_t.

Thanks!

---

> ### Author Response · Authors · 2019-11-06
> **Thanks for bringing this interesting work to our attention.**
>
> Thanks for your comments! We note that our work share similar idea with yours. Since this paper is submitted to NeurIPS2019 this May as well, it may be regarded as a concurrent work. We will mention and cite your work in our updated version of the paper.
>
> In table5, we don`t take the MFLOPs of coefficient prediction module and dynamic generation module into account because it is negligible. For example, when g_t=6, the computation cost of coefficient prediction module and dynamic generation module is only 3.37 and 2.75 MFLOPs respectively, while the computation cost of convolution is 129 MFLOPs.

---

### Public Comment · ~HC_Xu1 · 2020-10-28
**Could you open your code?**

this work looks easier to use, will you release your code in the future?

---

### Decision · Program_Chairs · 2019-12-19

**Decision:**

Reject

**Comment:**

The paper proposed the use of dynamic convolutional kernels as a way to reduce inference computation cost, which is a linear combination of static kernels and fused after training for inference to reduce computation cost. The authors evaluated the proposed methods on a variety models and shown good FLOPS reduction while maintaining accuracy.

The main concern for this paper is the limited novelty. There have been many works use dynamic convolutions as pointed out by all the reviewers. The most similar ones are SENet and soft conditional computation. Although the authors claim that soft conditional computation "focus on using more parameters to make models to be more expressive while we focus on reducing redundant calculations", the methods are pretty the same and moreover in the abstract of soft conditional computation they have "CondConv improves the performance and inference cost trade-off".